# Low-Rank Adapting Models for Sparse Autoencoders

**Matthew Chen** [* 1]   **Joshua Engels** [* 1]   **Max Tegmark** [1]

## Abstract

Sparse autoencoders (SAEs) decompose language model representations into a sparse set of linear latent vectors. Recent works have improved SAEs using language model gradients, but these techniques require many expensive backward passes during training and still cause a significant increase in cross entropy loss when SAE reconstructions are inserted into the model. In this work, we improve on these limitations by taking a fundamentally different approach: we use low-rank adaptation (LoRA) to finetune the *language model itself* around a previously trained SAE. We analyze our method across SAE sparsity, SAE width, language model size, LoRA rank, and model layer on the Gemma Scope family of SAEs. In these settings, our method reduces the cross entropy loss gap by 30% to 55% when SAEs are inserted during the forward pass. We also find that compared to end-to-end (e2e) SAEs, our approach achieves the same downstream cross entropy loss $3\times$ to $20\times$ faster on Gemma-2-2B and $2\times$ to $10\times$ faster on Llama-3.2-1B. We further show that our technique improves downstream metrics and can adapt multiple SAEs at once without harming general language model capabilities. Our results demonstrate that improving model interpretability is not limited to post-hoc SAE training; Pareto improvements can also be achieved by directly optimizing the model itself.[1]

## 1. Introduction

Although language models demonstrate profound capabilities in tasks such as in-context learning, mathematics, and coding (Brown et al., 2020; OpenAI, 2024; Team et al.,

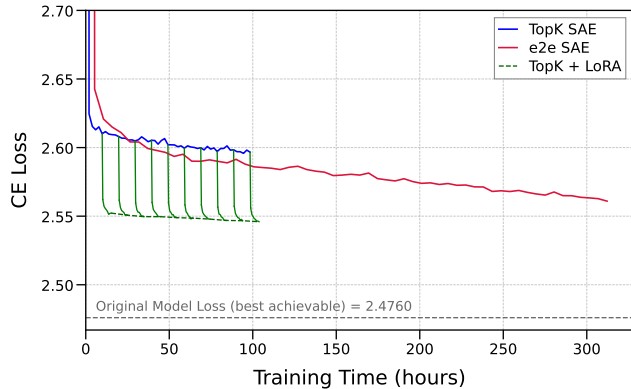

*Figure 1.* Cross entropy loss vs. training time over 2B tokens for Gemma-2-2B TopK SAEs with $\mathrm{width} = 18,432, \mathrm{L}_0 = 64$. We find that our method (TopK + LoRA in the plot) outperforms an e2e SAE and vanilla TopK SAE.

2023; Bubeck et al., 2023; Anthropic, 2024), the mechanisms behind these behaviors remain largely inscrutable. *Mechanistic interpretability* (MI) (Bereska & Gavves, 2024) seeks to understand these mechanisms by reverse engineering them into human understandable algorithms. In this work, we focus on better understanding features, the "variables" of model computation (Olah et al., 2020; Mueller et al., 2024; Marks et al., 2024).

A popular hypothesis in MI is the *Linear Representation Hypothesis* (LRH) (Elhage et al., 2022a; Park et al., 2023), which posits that features are one-dimensional directions in activation space. Although some recent research has called aspects of this hypothesis into question (Engels et al., 2024a; Csordás et al., 2024; Engels et al., 2024b), the LRH has been empirically validated for many language model features "in the wild" (Nanda et al., 2023; Heinzerling & Inui, 2024). Inspired by these successes, *sparse autoencoders* (SAEs) (Makhzani & Frey, 2013) have recently been applied to decompose language model hidden states into many linear features (Cunningham et al., 2023; Bricken et al., 2023). The latents SAEs learn are significantly more interpretable and monosemantic than the original neuron basis (Cunningham et al., 2023; Bricken et al., 2023).

While SAEs find interpretable latents, this comes at a cost: when SAE reconstructions are inserted back into the model and the forward pass is performed, the resulting cross en-

---

[*]Equal contribution [1]Massachusetts Institute of Technology, Cambridge, MA. Correspondence to: Matthew Chen <mattchen@mit.edu>.

*Proceedings of the $42^{nd}$ International Conference on Machine Learning*, Vancouver, Canada. PMLR 267, 2025. Copyright 2025 by the author(s).

[1]Code available at https://github.com/matchten/LoRA-Models-for-SAEs

tropy loss ($\mathcal{L}_{\text{SAE}}$) is significantly higher than the loss of the original model ($\mathcal{L}_{\text{BASE}}$). For example, when reconstructions from a TopK SAE are inserted into GPT-4, the resulting $\mathcal{L}_{\text{SAE}}$ is equivalent to the $\mathcal{L}_{\text{BASE}}$ of a model trained with just 10% of the pretraining compute of GPT-4 (Gao et al., 2024). Thus, previous work has extensively focused on optimizing SAE architectures to find Pareto improvements in the SAE sparsity vs. $\mathcal{L}_{\text{SAE}}$ frontier. This work includes TopK SAEs (Gao et al., 2024), Gated SAEs (Rajamanoharan et al., 2024a), JumpReLU SAEs (Rajamanoharan et al., 2024b), ProLU SAEs (Taggart, 2024), Switch SAEs (Mudide et al., 2024), and e2e SAEs (Braun et al., 2024).

However, an unexplored question is whether language models themselves can be optimized after SAE training to gain an additional Pareto improvement in sparsity vs. $\mathcal{L}_{\text{SAE}}$. In this work, we answer this question in the affirmative: we use *Low-Rank Adapters* (LoRA) (Hu et al., 2021) to reduce the KL divergence between the original model's logits and the model's logits with an SAE inserted. The resulting model-SAE combination improves in $\mathcal{L}_{\text{SAE}}$ and on a diverse set of downstream SAE metrics. Compared to existing proposals for training more interpretable models with SAEs (Lai & Heimersheim, 2024; Lai & Huang, 2024), we estimate our LoRA method is $10^7$ times faster [2]. Overall, we find that low-rank adapting models is a simple and efficient technique to improve the interpretability vs. performance trade-off.

Our contributions include the following:

1. To the best of our knowledge, we are the first to focus on improving the *model* around an existing SAE.
2. In Section 4.1, we analyze LoRA SAE training on the Gemma Scope (Lieberum et al., 2024) family of SAEs. Across SAE width, SAE sparsity, language model size, LoRA rank, and language model layer, we find a 30% to 55% improvement in $\mathcal{L}_{\text{SAE}}$–with final values between 0.01 and 0.17 nats–with especially large improvements in low sparsity regimes and larger models.
3. In Section 4.2, we compare our method to e2e SAEs on training time vs. $\mathcal{L}_{\text{SAE}}$ on Gemma-2-2B (Team et al., 2024) and Llama-3.2-1B (AI@Meta, 2024). We find that our method achieves the same $\mathcal{L}_{\text{SAE}}$ as e2e SAEs with between $2\times$ and $20\times$ less compute and $130\times$ fewer language model backward passes.
4. In Section 4.3, we perform LoRA SAE training with multiple SAEs inserted into Llama-3.1-8B (AI@Meta, 2024) and see large decreases in $\mathcal{L}_{\text{SAE}}$, demonstrating the potential of our technique for helping circuit analysis.
5. In Section 5, we show quantitative improvements on a diverse set of downstream tasks.
6. In Section 6, we find that we can achieve much of the

---

[2] Gemma-2-2B was trained with 6T tokens, whereas we use 15M tokens on up to 3% of the parameters; $6T/15M/0.03 = 1.3 \times 10^7$

benefit of full-parameter LoRA by training an adapter only on the layer after the SAE and that our adapters achieve most improvement on tokens with low $\mathcal{L}_{\text{SAE}}$.

## 2. Related Work

**SAE Architecture Improvements.** Early SAEs for language models used a simple linear encoder, ReLU activation with an $L_1$ penalty (approximating $L_0$), and a linear decoder (Bricken et al., 2023; Cunningham et al., 2023). The next generation introduced TopK and BatchTopK SAEs, which enforce sparsity by retaining only the $k$ largest activations (Gao et al., 2024; Bussmann et al., 2024), and GatedSAEs and JumpReluSAEs, which use gating functions and straight-through estimators to approximate direct $L_0$ optimization (Rajamanoharan et al., 2024a;b). These methods improve the $\mathcal{L}_{\text{SAE}}$ vs. sparsity tradeoff, though no single approach is definitively superior on downstream tasks (Karvonen et al., 2024). Beyond sparsity penalties, Braun et al. (2024) optimize SAEs for KL divergence with the model's logits to directly improve $\mathcal{L}_{\text{SAE}}$, while Olmo et al. (2024) incorporate model gradients into TopK activations for more causal representations. However, gradient-based methods introduce computational overhead and have a large limitation: SAEs are typically trained on cached activations without available gradients. (Lieberum et al., 2024).

**Fine-tuning SAEs.** While this paper focuses on fine-tuning a model around an SAE, another research direction explores fine-tuning SAEs. Some work tailors SAEs to specific domains by oversampling certain contexts (Bricken et al., 2024) or fine-tuning on domain-specific activations (Drori, 2024). Kissane et al. (2024) find that training SAEs on chat data captures the refusal latent, whereas training on the Pile (Gao et al., 2020) does not. Kutsyk et al. (2024) further analyze when base model SAEs generalize to a chat-tuned model, showing that it depends on the language model used.

**Training interpretable models.** We are aware of two prior works that investigate training more interpretable models using SAEs: both (Lai & Heimersheim, 2024) and (Lai & Huang, 2024) train SAEs and models concurrently, and find that this improves $\mathcal{L}_{\text{SAE}}$. However, because this requires training models from scratch, it is impractical to apply to existing models and is only shown to work in toy settings; in contrast, our method is extremely efficient, and we show it works on models up to 27B parameters. Many prior works also investigate this direction without SAEs. Elhage et al. (2022b) introduce the softmax linear unit (SoLU) activation function, which increases the fraction of interpretable neurons at no cost on downstream performance; Liu et al. (2023) propose a new loss term penalizing spatially distant connections in the network that leads to visually interpretable networks; Liu et al. (2024) introduce Kolmogorov-Arnold

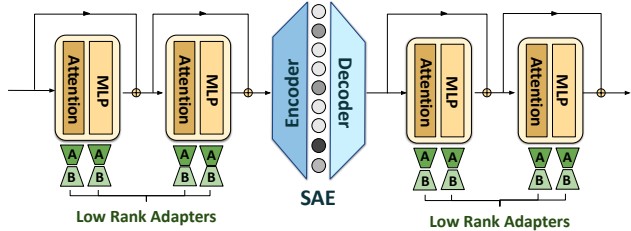

*Figure 2.* Visual representation of our method, with a local SAE trained on layer 12 and low-rank adapters trained on MLP and attention components on all layers.

Networks, an alternative to standard MLPs with trainable activation functions that can be replaced by symbolic formulas; and Heimersheim (2024) fine-tune out the LayerNorm components in GPT-2 with a small downstream loss effect.

**Parameter Efficient Fine Tuning.** *Parameter efficient fine tuning* (PEFT) reduces the cost of full supervised fine tuning by updating fewer effective parameters. One of the simplest and most effective PEFT methods is low-rank adaptation (LoRA) (Hu et al., 2021). LoRA works as follows: for a frozen pretrained weight matrix $W_0 \in \mathbb{R}^{d \times k}$, and low-rank matrices $A \in \mathbb{R}^{d \times r}$, $B \in \mathbb{R}^{r \times k}$ with $r \ll \min(d, k)$, the original forward pass $h(x) = W_0 x$ becomes

$$\hat{h}(\mathbf{x}) = W_0 x + ABx. \tag{1}$$

$A$ and $B$ can then be trained while the rest of the model is frozen, resulting in a low-rank update of the base model.

## 3. Optimizing Models for Sparse Autoencoders

In this section, we formally describe existing methods for training SAEs and our method of adapting models for SAEs. For a decoder only transformer with $L$ layers and hidden dimension $d$, input $\mathbf{x}_0$, and output $\mathbf{y}$, denote the activation after the $i$th layer by $\mathbf{x}_i$. Express the $i$th transformer block as a function $h_i$ such that the network can be expressed as

$$\mathbf{x}_i = h_i(\mathbf{x}_{i-1}) \qquad 1 \leq i \leq L \tag{2}$$
$$\mathbf{y} = \text{softmax}(\mathbf{x}_L) \tag{3}$$

### 3.1. TopK Sparse Autoencoders

SAEs learn an encoder $\mathbf{W}_{\text{enc}} \in \mathbb{R}^{m \times d}$ for $m \gg d$, a decoder $\mathbf{W}_{\text{dec}} \in \mathbb{R}^{d \times m}$ with unit norm columns, and biases $\mathbf{b}_{\text{enc}} \in \mathbb{R}^m, \mathbf{b}_{\text{dec}} \in \mathbb{R}^d$. We call the m columns of $\mathbf{W}_{\text{dec}}$ latents. For activation $\mathbf{x}_l$, the TopK SAE (Gao et al., 2024) reconstructs activation $\hat{\mathbf{x}}_l$ as follows:

$$\mathbf{z} = \text{TopK}(\mathbf{W}_{\text{enc}}(\mathbf{x}_l - \mathbf{b}_{\text{dec}}) + \mathbf{b}_{\text{enc}}) \tag{4}$$
$$\hat{\mathbf{x}}_l = \mathbf{W}_{\text{dec}} \mathbf{z} + \mathbf{b}_{\text{dec}} = \sum w_i \mathbf{f}_i \tag{5}$$

During training, the SAE minimizes the reconstruction error $\mathcal{L} = \|\mathbf{x}_l - \hat{\mathbf{x}}_l\|^2$. We train TopK SAEs with $k = 64$ for Gemma-2-2B and Llama-3.2-1B for 2B and 4B tokens, respectively, on the RedPajama dataset (Weber et al., 2024).

### 3.2. End-to-End Sparse Autoencoders

In an e2e SAE (Braun et al., 2024), the SAE minimizes KL divergence with the base model instead of reconstruction error. Formally, if we have

$$\hat{\mathbf{x}}_l = \text{SAE}(\mathbf{x}_l),$$
$$\hat{\mathbf{x}}_i = h_i(\hat{\mathbf{x}}_{i-1}) \qquad l < i \leq L,$$
$$\hat{\mathbf{y}} = \text{softmax}(\hat{\mathbf{x}}_L),$$

then the e2e SAE minimizes $\mathcal{L} = \text{KL}(\hat{\mathbf{y}}, \mathbf{y})$. For both e2e and TopK SAEs, we use a TopK activation function with the same sparsity to allow for fair comparisons.

### 3.3. JumpReLU Sparse Autoencoders

We also evaluate our method on the Gemma Scope JumpReLU SAEs. Instead of the TopK function, JumpReLU SAEs (Rajamanoharan et al., 2024b) use the JumpReLU activation function,

$$\text{JumpReLU}_\theta(z) := z H(z - \theta),$$

where $H$ is the Heaviside step function and $\theta > 0$ is the JumpReLU's threshold. The SAE is trained to minimize

$$\mathcal{L} = \|\hat{\mathbf{x}} - \mathbf{x}\|_2^2 + \lambda \|\mathbf{z}\|_0, \tag{6}$$

where $\mathbf{z}$ is defined from Equation (4).

### 3.4. Method for Low-Rank Adapting Models to SAEs

**Formulation.** We formally describe our method of optimizing models for SAEs, using notation from Equations (2)–(4). We insert a *frozen* SAE immediately after layer $\ell$, and the reconstructed activation $\hat{\mathbf{x}}_\ell = \text{SAE}(\mathbf{x}_\ell)$ propagates through the remaining layers to produce $\hat{\mathbf{x}}_i = h_i(\hat{\mathbf{x}}_{i-1})$ for $\ell + 1 \leq i \leq L$ and $\mathbf{y} = \text{softmax}(\mathbf{x}_L)$.

For JumpReLU SAEs we can only adapt layers *after* the SAE to ensure average sparsity is unaffected, while for TopK SAEs we can train adapters on all layers by maintaining the TopK constraint during training. We add low-rank adapters of rank $r$ in each MLP and attention sublayer of every layer we are adapting. Concretely, for each frozen weight matrix $W_i \in \mathbb{R}^{d_1 \times d_2}$, we add $A_i \in \mathbb{R}^{d_1 \times r}$ and $B_i \in \mathbb{R}^{r \times d_2}$ and modify the forward pass according to Equation (1).

We train only the low-rank adapters $\Theta = \{A_i\} \cup \{B_i\}$. For all experiments the training objective is the KL divergence between the next token probability distribution with and

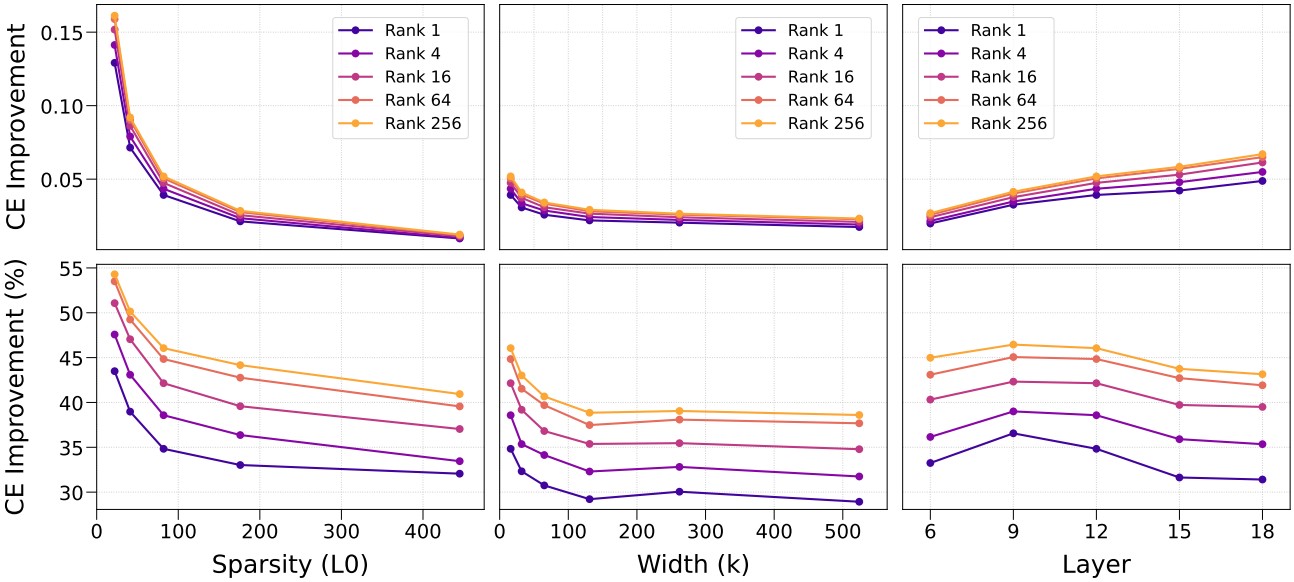

*Figure 3.* Cross entropy loss improvement (**Top**: absolute, **Bottom:** percentage of CE loss gap closed) using our method for Gemma Scope SAEs on Gemma-2-2B. **Left:** Scaling across sparsity with fixed width=16k and layer=12, we see the largest effect by percentage of our method at lower sparsities, but still substantial effect at higher sparsities as well. **Middle:** Scaling across width with fixed $L_0 = 68$ and layer=12, the highest effect by percentage is at low width but again this is not a large effect. **Right:** Scaling across layer with fixed $L_0 = 68$ and width=16k, the highest effect of our method by percentage is at layer 9 but it is mostly unaffected by layer.

without the SAE inserted:

$$\arg \min_{\Theta} \mathrm{KL}(\hat{\mathbf{y}}, \mathbf{y}) \tag{7}$$

By freezing both the SAE and the original model, this *parameter-efficient* method aligns the SAE-enhanced model to the behavior of the original model with minimal additional cost.

## 4. Results

In this section, we study how our method improves the cross entropy loss gap ($\mathcal{L}_{\mathrm{SAE}} - \mathcal{L}_{\mathrm{BASE}}$) before and after LoRA on a wide variety of SAEs and language models. Unless otherwise specified, we use a layer 12 residual stream SAE.

### 4.1. Scaling Laws for Downstream Loss

We first explore the scaling behavior of low-rank adapting models to SAEs across SAE sparsity, SAE width, language model size, LoRA rank, and model layer. Specifically, we use Gemma Scope's JumpReLU SAEs (Rajamanoharan et al., 2024b). To ensure we do not affect the average sparsity of these JumpReLU SAEs, we only finetune the layers *after* the SAE. Over different sparsities, widths, and layers, we track the absolute and percent improvement in $\mathcal{L}_{\mathrm{SAE}} - \mathcal{L}_{\mathrm{BASE}}$ after low-rank fine-tuning. We train on $15M$ random tokens of The Pile (uncopyrighted) dataset (Gao

et al., 2020), and evaluate on a held out validation set of $1M$ random tokens. We report our findings in Figure 4 for model size and in Figure 3 for other scaling axes.

We find that across all of the scaling regimes we test, we close the $\mathcal{L}_{\mathrm{SAE}} - \mathcal{L}_{\mathrm{BASE}}$ gap by at least 30%, and sometimes by up to 55%. We find that using larger rank LoRA adapters reliably decreases the final $\mathcal{L}_{\mathrm{SAE}}$; this, combined with the fact that we adapt on only 15M tokens and do not see our adapters finish converging, implies that with more compute our method may be even more successful.

We find that over varying layers, the improvement is largest for middle layers, although this result is not extremely strong (according to e.g. (Lad et al., 2024), this may arise from the fact that middle layers tend to have richer and more expressive representations that local SAEs may struggle reconstructing). We also find that the improvement is largest on lower sparsities, lower widths, and larger models; all of these results may be caused by these SAEs having a higher cross entropy loss gap to start with. We do still find it extremely promising that the effectiveness of our technique increases on larger models.

### 4.2. Downstream Loss vs. Computational Cost

Next, we study the frontier of $\mathcal{L}_{\mathrm{SAE}}$ versus training time for TopK SAEs, e2e SAEs, and low-rank adapted TopK SAEs. To do this, we need to train our own TopK and e2e SAEs to

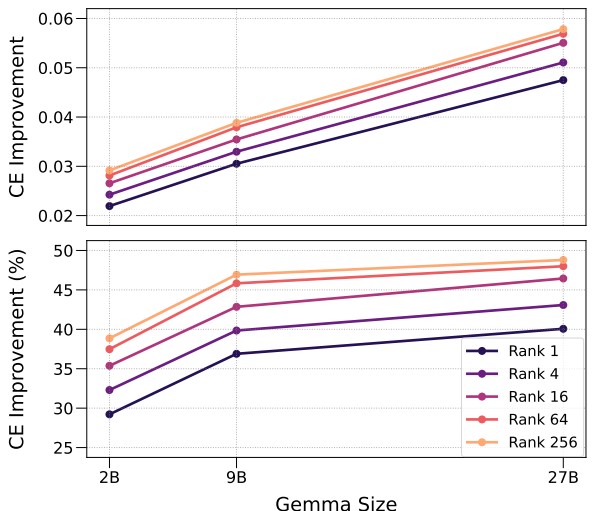

*Figure 4.* Cross entropy loss improvement (**Top**: absolute, **Bottom:** percentage) for Gemma Scope SAEs of width $16k$ and $L_0$ closest to 70 on Gemma-2-2B, 7B, and 27B. We find that our method works increasingly well on larger models.

*Table 1.* Gemma-2-2B timing results and speedups, nearest hour

| CE Loss | TopK + LoRA | TopK | e2e | Speedup |
|---------|-------------|------|------|---------|
| 2.60 | **12h** | 59h | 37h | **3.05x** |
| 2.59 | **12h** | — | 79h | **6.53x** |
| 2.58 | **12h** | — | 148h | **12.18x** |
| 2.57 | **12h** | — | 243h | **20.01x** |
| 2.55-2.57 | **12h–107h** | — | — | — |

*Table 2.* Llama-3.2-1B timing results and speedups, nearest hour

| CE Loss | TopK + LoRA | TopK | e2e | Speedup |
|---------|-------------|------|------|---------|
| 2.73 | **9h** | — | 96h | **10.38x** |
| 2.72 | **12h** | — | 113h | **9.08x** |
| 2.71 | **19h** | — | 135h | **7.14x** |
| 2.70 | **70h** | — | 156h | **2.23x** |
| 2.67-2.70 | — | — | **156h–213h** | — |

get their training curves. We also use checkpoints from the TopK training run to get the training curve for TopK SAEs + LoRA; after every 10% training checkpoint of the TopK SAEs, we low-rank adapt the model checkpoint with rank 64 LoRA on all layers.

We train on layer 12 of Llama-3.2-1B and Gemma-2-2B. We train TopK and e2e SAEs for 4B tokens on Llama-3.2-1B and for 2B tokens on Gemma-2-2B (similar to the number of tokens trained on for Gemma Scope SAEs). On each TopK SAE training checkpoint of Llama-3.2-1B we do LoRA finetuning for 100M tokens, while we finetune for 15M tokens on Gemma-2-2B TopK SAE checkpoints.

We show the Pareto cross entropy frontiers for Gemma-2-2B and Llama-3.2-1B in Figures 1 and 5, respectively, where our method clearly dominates. Quantitatively, we show the speedup in wall clock time in achieving various cross entropy loss threshold when using TopK + LoRA versus e2e in Tables 1 and 2; our speedups ranging from 2× to 20×. Our approach (TopK + LoRA) also performs 130× fewer backward passes through the model than e2e SAEs on Gemma-2-2B and 40× fewer backward passes on Llama-3.2-1B. Finally, we do note, however, that e2e SAEs achieve a lower final CE loss than our method on Llama-3.2-1B (although not on Gemma-2-2B).

### 4.3. Adapting Multiple SAEs

Inserting multiple SAEs at once into a language model causes $\mathcal{L}_{SAE}$ to grow extremely rapidly (e.g. as shown in Figure 6, we find that inserting 5 SAEs leads to a cross entropy error of almost 10 nats, which is worse than a uni-

gram model (Gao et al., 2024)). At the same time, inserting multiple SAEs at once is a very useful task for circuit analysis, since it allows one to determine dependencies between SAE latents (in practice, past SAE circuits work (Marks et al., 2024) has used error terms to overcome this limitation, which results in less interpretable circuits). Thus, we adapt our procedure to work with multiple SAEs: we insert all SAEs at once during training, and otherwise follow Section 3.4. We measure the performance of our technique with the "Compound Cross Entropy Loss" (Lai & Heimersheim, 2024), which is simply $\mathcal{L}_{SAE}$ with all SAEs inserted.

We use the Llama Scope (He et al., 2024) set of SAEs (width $= 131,072, L_0 = 50$) trained on Llama-3.1-8B. Because these are TopK SAEs, we can train the LoRA layers without worrying about violating sparsity constraints. We train with the following configurations of SAEs, chosen to maximize the distance between adjacent pairs of SAEs: 1 SAE at layers $\{16\}$; 3 SAEs at layers $\{10, 20, 30\}$; 5 SAEs at layers $\{6, 12, 18, 24, 30\}$; 7 SAEs at layers $\{4, 8, 12, \dots, 28\}$; 10 SAEs at layers $\{3, 6, 9, \dots, 30\}$; and 15 SAEs at layers $\{2, 4, 6, \dots, 30\}$.

Our results (see Figure 6) show that this method significantly reduces compound CE loss; for example, using LoRA, the compound CE loss for 7 SAEs goes from 7.83 nats to 2.78 nats, while the compound CE loss for 3 SAEs goes from 3.55 nats to 2.45 nats (which is lower than the original validation CE loss with a *single* SAE and no LoRA). Thus, our technique seems extremely promising for circuit analysis.

## 5. Downstream Improvements

A common criticism of previous SAE optimizations is the lack of grounded metrics for evaluating how good an SAE

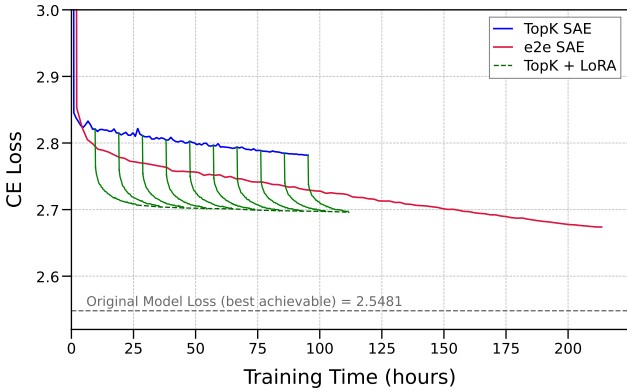

Figure 5. Cross entropy loss vs. training time for Llama-3.2-1B with TopK SAEs of $L_0 = 64$ and width 16384. Our method (TopK + LoRA) achieves lower CE loss sooner than e2e SAE or vanilla TopK SAEs

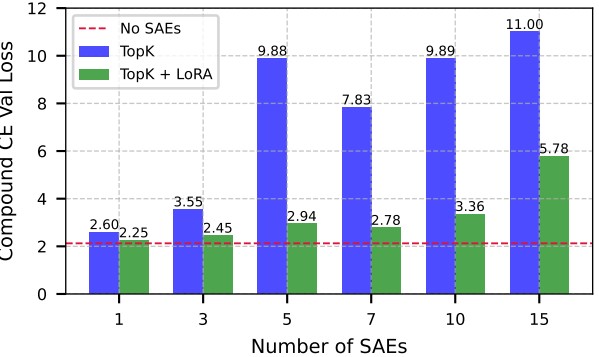

Figure 6. Downstream cross entropy loss when multiple Llama Scope SAEs are inserted into Llama-3.1-8B at once. "Base" is the original loss without any fine-tuning, while "LoRA" is the loss after 15M tokens of LoRA training.

is. Prior work has largely relied on unsupervised metrics such as in Section 4. Recent work, however, has introduced evaluation metrics to measure a model and SAE according to their performance on downstream tasks (Karvonen et al., 2024; Pres et al., 2024). Thus, in this section, we evaluate our method on a diverse set of downstream benchmarks:

1. In Section 5.1, we show that using LoRA on all layers improves downstream tasks on SAEBench.
2. In Section 5.2, we introduce a novel steering metric and show that our method improves on it. We introduce a new metric because SAEBench metrics do not test the effects of SAEs on next token prediction.
3. In Section 5.3, we show that our method improves overall model performance with the SAE inserted on MMLU, HellaSwag, and TruthfulQA.

## 5.1. SAEBench

To address the core challenge of measuring how effectively a model and SAE work together, Karvonen et al. (2024) introduce SAEBench, a benchmark of SAE metrics that are faithful to possible real world use cases. For the Gemma-2-2B TopK SAE ($L_0 = 64$) we trained in Section 4.2, we evaluate the model with the SAE and the model with the SAE + LoRA on SAEBench.

Specifically, we look at spurious correlation removal (SCR), targeted probe perturbation (TPP), SPARSE PROBING, automated interpretability (AUTOINTERP), and feature absorption (ABSORPTION). SCR measures the separation of latents for different concepts, with higher scores indicating better ability to debias a classifier. TPP evaluates the impact of ablating specific latents on probe accuracy, where higher scores reflect well-isolated latents corresponding to classes on a dataset. SPARSE PROBING tests the accuracy

Table 3. Using the same TopK SAE trained on Gemma-2-2B, we compare the SAEBench metrics when the underlying model is low-rank adapted with rank 64. We see across most applicable metrics, the LoRA model shows meaningful improvement. Full results over various thresholds are displayed in Table 6.

| DOWNSTREAM METRIC | TOPK + LORA | TOPK |
|---|---|---|
| SCR (MAX) | **0.526** | 0.448 |
| SCR (AVERAGE) | **0.314** | 0.289 |
| TPP (MAX) | **0.412** | 0.372 |
| TPP (AVERAGE) | **0.145** | 0.111 |
| SPARSE PROBING (TOP 1) | **0.760** | 0.732 |
| SPARSE PROBING (TEST) | **0.956** | 0.955 |
| AUTOINTERP | 0.830 | **0.832** |
| ABSORPTION | 0.210 | **0.205** |

of a k-sparse probe trained on SAE latents, with higher scores indicating better feature learning. AUTOINTERP, assessed using an LLM judge (GPT-4o-mini (OpenAI, 2024)), quantifies the interpretability of SAE latents. ABSORPTION quantifies to what extent latents are "absorbed" together to improve sparsity. All metrics range from 0 to 1, with higher being better except for ABSORPTION. [3]

We display our results in Table 3, showing our low-rank adapted model outperforms the base model on TPP, SCR, and SPARSE PROBING, while very slightly underperforming on AUTOINTERP and ABSORPTION.

## 5.2. Feature Steering

In this section, we demonstrate that the LoRA tuned model improves at activation steering–repressing or eliciting model

---

[3]Excluded from our results are the RAVEL and UNLEARNING SAEBench metrics. RAVEL is not yet implemented in SAEBench and UNLEARNING is recommended for instruct models only.

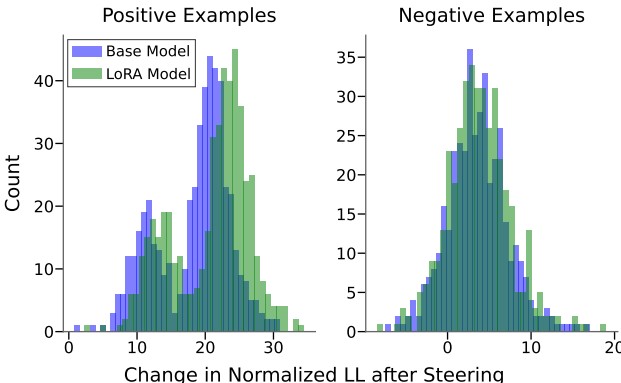

*Figure 7.* The distribution of normalized log-likelihood change post-steering is more pronounced on positive examples for the LoRA model. Results shown for the SAE latent responsible for "Donald Trump".

behavior by scaling a steering vector–using its SAE latents. Given an SAE latent $\mathbf{v} \in \mathbb{R}^d$ at layer $l$, we steer via

$$\mathbf{x}_l \rightarrow \mathbf{x}_l + \alpha(\mathbf{x}_l \cdot \mathbf{v})\mathbf{v}, \quad \alpha \in \mathbb{R}. \qquad (8)$$

We assess steering effectiveness following (Pres et al., 2024), who evaluate steering by analyzing increases in likelihood for "positive" texts (aligned with the desired behavior) and decreases for "negative" texts (not aligned). We note that Olmo et al. (2024) also introduce an SAE steering evaluation, but it does not follow the best practices for steering laid out by (Pres et al., 2024) so we do not use it.

Our method is slightly different than (Pres et al., 2024) because we are comparing different *models* with the same steering method instead of comparing different steering methods on the same model. For a given SAE latent, we steer on a dataset of 500 positive and negative samples. The negative dataset consists of an equal mix of arabic tweets (Pain, 2024), medical facts (MedAlpaca, 2024), recipes (Corbt, 2024), shakespearean quotes (Roudranil, 2024), and law texts (GPT-4o-mini generated). We generate the positive datasets by selecting a latent about 1) machine learning, 2) San Fransisco, 3) Donald Trump, and 4) Covid-19. We then generate text samples where that latent fires using GPT-4o-mini. See Appendix A.1.2 for full prompt details.

Following (Pres et al., 2024), we compute mean token log-likelihoods before and after steering, normalizing them so the original likelihoods span 0 to 100. We tune the hyperparameter $\alpha$ in Equation (8) by selecting a value that increases the likelihood of positive samples while minimizing likelihood increases on a validation subset of negative samples (medical facts). After tuning, we evaluate the effect of $\alpha$ on a test consisting of the remaining negative samples. This tuning process is repeated for the base and LoRA models.

*Table 4.* For each SAE latent, $\Delta_{\text{POSITIVE}}$ and $\Delta_{\text{NEGATIVE}}$ denote the 90% CI improvement in normalized log-likelihood increase when using the LoRA model for steering on positive and negative examples, respectively. Because $\Delta_{\text{POSITIVE}} > 0$ and $\Delta_{\text{NEGATIVE}} \leq 0$, we see the LoRA model is better at steering for a given SAE latent while not affecting other features.

| SAE FEATURE | $\Delta_{\text{POSITIVE}}$ | $\Delta_{\text{NEGATIVE}}$ |
|---|---|---|
| MACHINE LEARNING | $\mathbf{0.86 \pm 0.82}$ | $-0.84 \pm 0.43$ |
| SAN FRANCISCO | $\mathbf{0.97 \pm 0.76}$ | $0.06 \pm 0.20$ |
| DONALD TRUMP | $\mathbf{2.50 \pm 0.56}$ | $0.20 \pm 0.40$ |
| COVID-19 | $\mathbf{0.44 \pm 0.25}$ | $-0.01 \pm 0.06$ |

To compare models, let $\Delta_{\text{POSITIVE}}$ and $\Delta_{\text{NEGATIVE}}$ represent the change in normalized likelihoods for positive and negative datasets when switching from the base to the LoRA model. The LoRA model is better suited for steering if $\Delta_{\text{POSITIVE}} > 0$ and $\Delta_{\text{NEGATIVE}} \leq 0$. We compute 90% confidence intervals for $\Delta_{\text{POSITIVE}}$ and $\Delta_{\text{NEGATIVE}}$ across 500 examples for each of our four SAE latents. Results are summarized in Table 4. We show a histogram of changes across all examples after steering for the best performing latent, "Donald Trump", in Figure 7.

### 5.3. General Language Model Capabilities

In addition to downstream tasks, we evaluate the model's general language capabilities on MMLU (Hendrycks et al., 2020), HellaSwag (Zellers et al., 2019), and TruthfulQA (Lin et al., 2021) in two different regimes: comparing evals when the SAE is inserted and when the SAE is not inserted. In other words, the four settings are: 1) **SAE**, the original model with the SAE, and 2) **SAE + LoRA**, a low rank adapted model with the SAE, 3) **Original**, the original model with no SAE, 4) **LoRA**, the adapted model with no SAE. Our results, shown in Table 5, show that across all Gemma model sizes and across benchmarks, our adapted regime is not hurt capability wise and even frequently *outperforms* the original model. In other words, adapting the model to be more faithful to its SAE latents does *not* harm general language model capability.

## 6. Analyzing Why Our Method Works

### 6.1. Per Token Improvement Breakdown

In this experiment, we analyze how LoRA impacts $\mathcal{L}_{\text{SAE}}$ improvements. Figure 8 shows the distribution of $\Delta\mathcal{L}_{\text{SAE}}$ between the original and LoRA models across 15M validation tokens. The per-token loss change varies greatly, with the loss on 37% of tokens even getting worse (see the degradation histogram in the figure). Most of the overall improvement comes from small per-token decreases in loss (roughly $10^{-2}$ to 1 nats), suggesting LoRA improves loss

*Table 5.* Comparisons of original model performance to performance with the SAE inserted, the SAE inserted with LoRA, and just LoRA. Error ranges represent one standard error; largest value between non-adapted and adapted versions are bolded. Note that even without the SAE the LoRA model is frequently better; thus, the LoRA adapter we train for the SAE does not harm general model performance.

| | GEMMA-2-2B | | | |
| --- | --- | --- | --- | --- |
| **METRIC** | SAE | SAE + LoRA | ORIGINAL | LoRA |
| MMLU | $44.2 \pm 0.4$ | $\mathbf{45.8 \pm 0.4}$ | $49.3 \pm 0.4$ | $\mathbf{50.0 \pm 0.4}$ |
| HELLASWAG | $50.9 \pm 0.5$ | $\mathbf{52.1 \pm 0.5}$ | $55.0 \pm 0.5$ | $\mathbf{56.0 \pm 0.5}$ |
| BLEU | $29.9 \pm 1.6$ | $\mathbf{30.6 \pm 1.6}$ | $30.4 \pm 1.6$ | $\mathbf{32.4 \pm 1.6}$ |
| ROUGE-1 | $28.2 \pm 1.6$ | $\mathbf{28.5 \pm 1.6}$ | $26.9 \pm 1.6$ | $\mathbf{30.2 \pm 1.6}$ |
| ROUGE-2 | $24.8 \pm 1.5$ | $\mathbf{26.6 \pm 1.5}$ | $25.6 \pm 1.5$ | $\mathbf{29.1 \pm 1.6}$ |
| MC1 | $23.1 \pm 1.5$ | $\mathbf{23.4 \pm 1.5}$ | $24.1 \pm 1.5$ | $\mathbf{24.3 \pm 1.5}$ |
| | GEMMA-2-9B | | | |
| **METRIC** | SAE | SAE + LoRA | ORIGINAL | LoRA |
| MMLU | $64.2 \pm 0.4$ | $\mathbf{65.7 \pm 0.4}$ | $\mathbf{70.0 \pm 0.4}$ | $68.8 \pm 0.4$ |
| HELLASWAG | $58.3 \pm 0.5$ | $\mathbf{59.6 \pm 0.5}$ | $61.2 \pm 0.5$ | $\mathbf{61.9 \pm 0.5}$ |
| BLEU | $40.9 \pm 1.7$ | $\mathbf{42.4 \pm 1.7}$ | $\mathbf{43.8 \pm 1.7}$ | $43.6 \pm 1.7$ |
| ROUGE-1 | $39.0 \pm 1.7$ | $\mathbf{40.6 \pm 1.7}$ | $42.7 \pm 1.7$ | $\mathbf{43.5 \pm 1.7}$ |
| ROUGE-2 | $33.4 \pm 1.7$ | $\mathbf{36.4 \pm 1.7}$ | $38.3 \pm 1.7$ | $\mathbf{38.8 \pm 1.7}$ |
| MC1 | $27.1 \pm 1.6$ | $\mathbf{28.0 \pm 1.6}$ | $30.5 \pm 1.6$ | $\mathbf{31.0 \pm 1.6}$ |
| | GEMMA-2-27B | | | |
| **METRIC** | SAE | SAE + LoRA | ORIGINAL | LoRA |
| MMLU | $70.9 \pm 0.4$ | $\mathbf{71.3 \pm 0.4}$ | $72.1 \pm 0.3$ | $\mathbf{72.7 \pm 0.3}$ |
| HELLASWAG | $61.0 \pm 0.5$ | $\mathbf{62.7 \pm 0.5}$ | $65.3 \pm 0.5$ | $\mathbf{65.5 \pm 0.5}$ |
| BLEU | $\mathbf{40.9 \pm 1.7}$ | $38.9 \pm 1.7$ | $41.1 \pm 1.7$ | $\mathbf{41.9 \pm 1.7}$ |
| ROUGE-1 | $\mathbf{41.0 \pm 1.7}$ | $38.3 \pm 1.7$ | $40.9 \pm 1.7$ | $\mathbf{41.7 \pm 1.7}$ |
| ROUGE-2 | $\mathbf{37.1 \pm 1.7}$ | $35.3 \pm 1.7$ | $36.2 \pm 1.7$ | $\mathbf{37.1 \pm 1.7}$ |
| MC1 | $30.2 \pm 1.6$ | $\mathbf{31.5 \pm 1.6}$ | $\mathbf{33.8 \pm 1.7}$ | $32.9 \pm 1.6$ |

across many tokens rather than a more bimodal distribution.

## 6.2. Activation Distances

One concern identified by (Braun et al., 2024) is that optimizing towards KL divergence may lead the activations to be off distribution and follow a different computational path through the model. We find this is *not* the case with our method: as shown in Figure 11, over a validation set of 500k tokens, our method slightly *decreases* the distance between activations after the SAE and activations in the original model, while the cosine similarities slightly *increase*. In other words, the adapted model + SAE follows a closer computation path to the original modle than the original model + SAE.

## 6.3. Single Layer Adapters

Another question we are interested in is which LoRA layers are most important for reducing $\mathcal{L}_{\text{SAE}}$. In Figure 9, we plot the results of an experiment where we train LoRA adapters on each individual layer after the layer with the inserted SAE. We find that the LoRA performance degrades as it gets farther from the original layer. Interestingly, we also find that training LoRA adapters on just the first layer after the SAE achieves 88.14% of the loss reduction in training LoRA adapters on *all* the layers after the SAE, suggesting the loss improvement mechanism may be reasonably simple.

## 7. Conclusion

Low-rank adapting models for SAEs provides a fast, cheap, and effective path to producing interpretable combined model and SAE systems. Moreover, low-rank adapted mod-

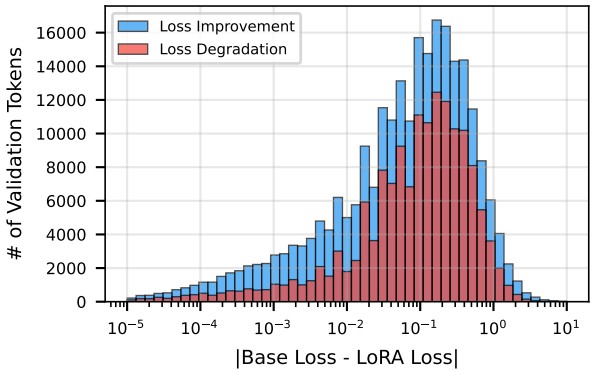

*Figure 8.* Distribution of loss improvements and loss degredations across validation tokens. We see that more tokens have a loss improvement than degredation (although a substantial number have a degradation) and most loss improvements and degredations happen in a range of about 0.01 to 1 $\Delta \mathcal{L}_{\text{SAE}}$ nats.

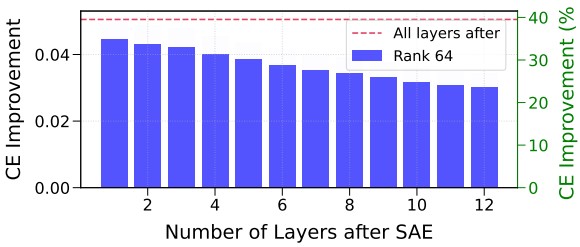

*Figure 9.* Plot of $\mathcal{L}_{\text{SAE}}$ when running LoRA on just a single layer of Gemma Scope 2B. We find that LoRA on layers closer to the SAE layer do better, and that also LoRA on just the next layer achieves much of the loss reduction of training on all layers.

els are "better" at using their SAEs for downstream tasks. Crucially, our work challenges the prevailing assumption that improving interpretability must rely solely on post-hoc model decomposition. We hypothesize that our method is much faster than e2e training because modifying the entire model gives many more degrees of freedom for the optimization procedure to work with; thus, our work suggests that focusing more on the larger space of possible language model modifications may be fruitful. We hope the results in this paper lay the groundwork for further such techniques.

### 7.1. Limitations

Mechanistic interpretability work usually interprets a frozen model, while in our work we change it; for some applications, this might not be acceptable. However, we do note that we use LoRA to decrease the KL divergence with the original model, so if one is using SAEs anyways, our method creates a more faithful model. Additionally, we do not yet show that our technique helps discover more faithful circuits for model behaviors; we leave this important direction for future work. Another limitation is that it seems the e2e

SAEs may not have finished converging in our experiments, so we cannot compare converged accuracy; however, we had already trained for more than a week before stopping, so training to full convergence may not be practical. We use the learning rates suggested in Gao et al. (2024), but it is possible that further tuning of the learning rate could make the e2e SAE train faster; for reasons of compute limitations, we do not experiment with learning rate. Finally, we note that if we had used LoRA for more tokens we may have gotten an additional improvement on the Gemma Scope scaling experiments; however, our results are still quite strong, and the fact that they were achieved with just 15M tokens shows the efficiency of our technique.

## Impact Statement

We believe that increasingly powerful language models and other AI systems pose many safety risks (e.g. deception, power seeking, misinformation, bias, CBRN risks; see (Slattery et al., 2024) for a complete summary). MI and other fields that try to better understand LLMs are motivated by reducing these risks (see (Bereska & Gavves, 2024) and (Sharkey et al., 2025) for more in depth reviews of MI and AI safety and discussions of open problems). Thus, because the goal in our work is to train more interpretable systems at a fixed level of fidelity to the original (uninterpretable) model, we do not foresee negative consequences of our work; on the contrary, we believe our work is broadly beneficial for AI safety.

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

# A. Appendix

## A.1. Steering

### A.1.1. DISTRIBUTION PLOTS

In Figure 10, we plot histograms for the changes in normalized log-likelihoods for each of the four datasets from Table 4.

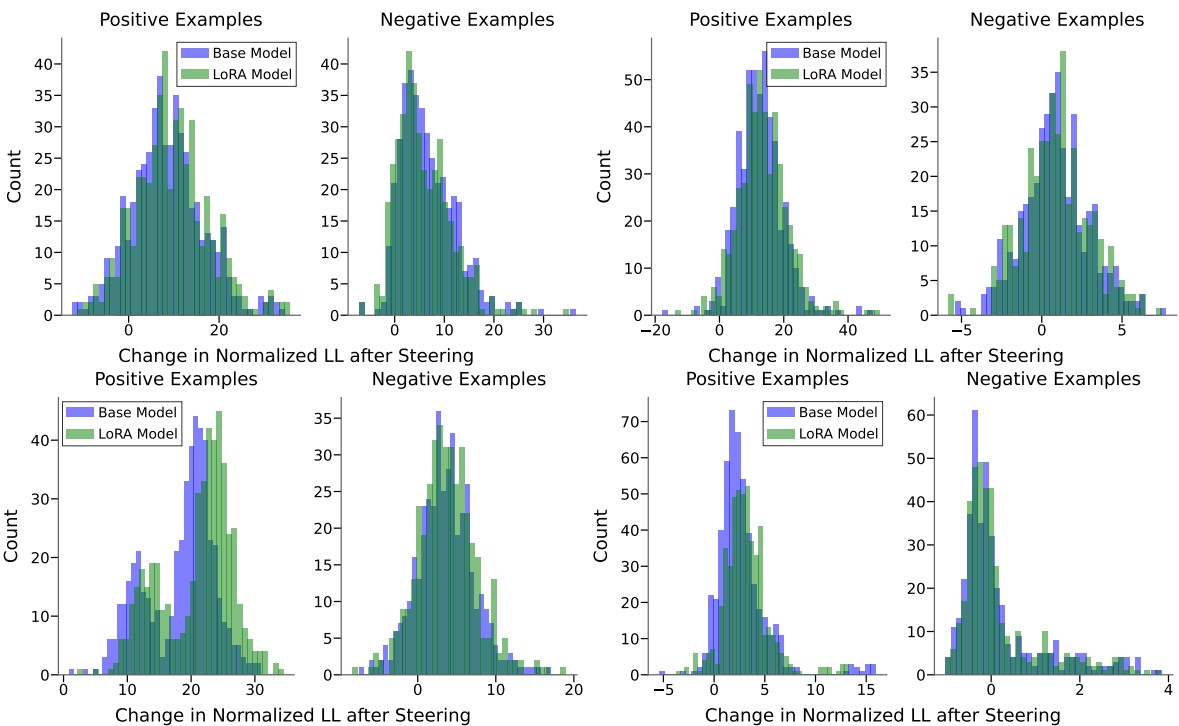

*Figure 10.* Distribution plots of the change in normalized log-likelihood after steering for various SAE latents. **Top left** is for machine learning (neuron 8421). **Top right** is for San Francisco (neuron 2195). **Bottom left** is for Donald Trump (neuron 13677). **Bottom right** is for COVID-19 (neuron 17811).

### A.1.2. DATASETS

To generate our "positive" examples dataset, we generate examples eliciting the SAE feature with GPT-4o-mini. We prompt using the following chat template.

```
prompt = """
    Generate {num_examples} text examples that have the following feature:
    {feature_description}

    Below are examples of text that have the feature described above.

    Examples:
    {examples}

    Each text example should be around **twelve** words long and be unique.
    Try to be varied in the content of the examples.
"""
```

## A.2. SAEBench Metrics

Here, we define in more detail the SAEBench metrics defined by (Karvonen et al., 2024) and used in Section 5.1, along with the full results over different hyperparameter splits.

**Absorption.** Feature absorption occurs when a latent representing concept $A$ implicitly encodes a related concept $B$ (e.g., *Elephant* $\Rightarrow$ *gray*), leading to redundancy or loss of interpretability. This phenomenon disrupts feature disentanglement, as absorbed features may activate inconsistently, obscuring their semantic meaning.

To measure absorption, (Karvonen et al., 2024) adapt the method of (Chanin et al., 2024) using a first-letter classification task. A logistic regression probe is trained on residual stream activations to establish a ground-truth feature direction. They then perform $k$-sparse probing on SAE latents, identifying primary latents responsible for the task. If increasing $k$ significantly improves F1 by some threshold, the new latent is classified as a feature split.

They then detect absorption by identifying test cases where primary latents fail while the probe succeeds. A latent is flagged as absorbing the feature if it strongly aligns with the probe in cosine similarity and accounts for a sufficient fraction of the probe projection.

**Spurious Correlation Removal.** The spurious correlation removal (SCR) metric evaluates whether the SAE captures separate latents for distinct concepts (e.g., gender vs. profession). A classifier is trained on a deliberately "biased" dataset (e.g., only male + professor, female + nurse), thereby picking up the spurious correlation, and then the latents most associated with the spurious feature (e.g., gender) are zero-ablated.

During evaluation, the classifier is to be debiased. Choosing the top $n$ latents according to their probe attribution score, a modified classifier is defined in which all latents except for the spuriously correlated latent are zero ablated. Evaluated on a balanced dataset, this modified classifier's accuracy in classifying its concept is tracked, and the metric is defined as

$$S_{\text{SHIFT}} = \frac{A_{\text{abl}} - A_{\text{base}}}{A_{\text{oracle}} - A_{\text{base}}},$$

where $A_{\text{abl}}$ is the probe accuracy after ablation, $A_{\text{base}}$ is the original spurious probe's accuracy, and $A_{\text{oracle}}$ is the accuracy of a probe directly trained on the concept. This SHIFT score quantifies how much ablation improves accuracy (removing the spurious signal), relative. A higher score indicates better separation of the spurious feature and stronger debiasing.

**Targeted Probe Perturbation.** SHIFT operates on datasets with correlated labels. To extend SHIFT to all multiclass NLP datasets, (Karvonen et al., 2024) introduce TPP, a method that identifies structured sets of SAE latents that disentangle dataset classes. This approach involves training probes on model activations and assessing the impact of ablating specific latent sets on probe accuracy. Ideally, removing a disentangled set of latents should only impact the corresponding class probe while leaving other class probes unaffected.

Consider a dataset where each input is assigned a single label from a set of $m$ possible concepts, $\mathcal{C} = \{c_1, c_2, ..., c_m\}$. For each class indexed by $i \in \{1, ..., m\}$, the most relevant latents $L_i$ are determined using probe attribution scores. To evaluate their effect, the dataset is partitioned into instances belonging to the target concept $c_i$ and a mixed subset containing randomly sampled instances from other labels.

A linear classifier $C_j$ is defined to predict concept $c_j$ with an accuracy of $A_j$. Furthermore, let $C_{i,j}$ denote the classifier for $c_j$ when latents in $L_i$ are ablated. The accuracy of each classifier $C_{i,j}$ on the corresponding dataset partition for $c_j$ is then computed as $A_{i,j}$. The TPP metric is given by:

$$S_{\text{TPP}} = \frac{1}{m} \sum_{i=j} (A_{i,j} - A_j) - \frac{1}{m} \sum_{i \neq j} (A_{i,j} - A_j)$$

This metric quantifies the extent to which ablating a disentangled set of latents selectively affects its corresponding class. A well-disentangled latent representation should cause a significant accuracy drop when $i = j$ (i.e., ablating latents relevant to class $i$ in classifier $C_i$) while having minimal effect when $i \neq j$.

**Sparse Probing.** To evaluate the SAE's ability to learn specific features, SAEs are tested on diverse tasks (e.g., language ID, profession classification, sentiment analysis). Inputs are encoded with the SAE, mean-pooled over non-padding tokens, and the top-K latents are selected via maximum mean difference. A logistic regression probe is trained on these latents and

*Table 6.* Using the same TopK SAE trained on Gemma-2-2B, we compare the SAEBench metrics when the underlying model is low-rank adapted with rank 64. The threshold hyperparameter for SCR and TPP denotes how many of the top $n$ latents are used in the modified classifier.

| Downstream Metrics | LoRA Model | Base Model |
|---|---|---|
| SCR Metric @2 | 0.094 | **0.097** |
| SCR Metric @5 | **0.196** | 0.177 |
| SCR Metric @10 | **0.260** | 0.253 |
| SCR Metric @20 | **0.336** | 0.327 |
| SCR Metric @50 | 0.447 | **0.448** |
| SCR Metric @100 | **0.526** | 0.400 |
| SCR Metric @500 | **0.342** | 0.325 |
| | | |
| TPP Metric @2 | **0.013** | 0.007 |
| TPP Metric @5 | **0.023** | 0.014 |
| TPP Metric @10 | **0.035** | 0.023 |
| TPP Metric @20 | **0.085** | 0.039 |
| TPP Metric @50 | **0.184** | 0.128 |
| TPP Metric @100 | **0.266** | 0.194 |
| TPP Metric @500 | **0.412** | 0.372 |
| | | |
| Sparse Probing (Top 1) | **0.760** | 0.732 |
| Sparse Probing (Top 2) | **0.833** | 0.832 |
| Sparse Probing (Top 5) | 0.875 | 0.875 |
| Sparse Probing (Top 10) | **0.910** | 0.907 |
| Sparse Probing (Top 20) | 0.930 | 0.930 |
| Sparse Probing (Top 50) | 0.946 | 0.946 |
| Sparse Probing (Test) | **0.956** | 0.955 |
| | | |
| Autointerp | 0.830 | **0.832** |
| Absorption | 0.210 | **0.205** |

evaluated on a held-out test set to assess how well the SAE captures the target features. A higher score reflects better feature representation (Karvonen et al., 2024).

### A.3. Activation Distances

In Figure 11 we show how low rank adapting the model affects the cosine similarity and $L_2$ distance of the model activations with and without the SAE.

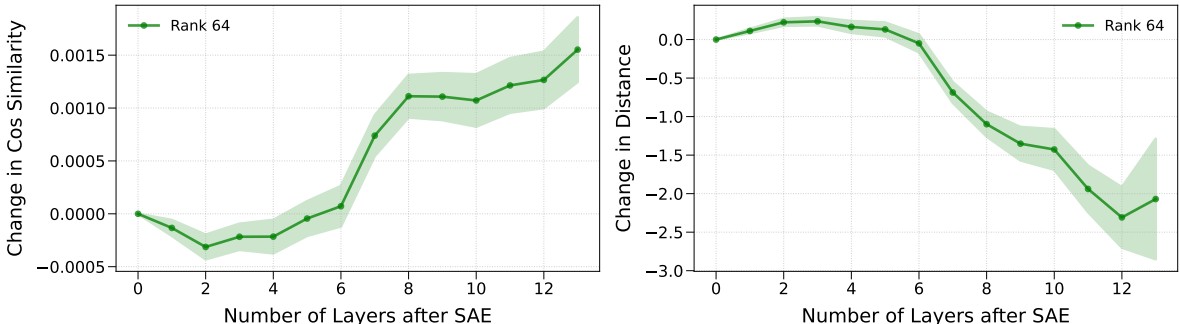

*Figure 11.* Change in average distance to original model activations before and after applying LoRA; increases in cosine similarity (**Left**) and decreases in Euclidean distance (**Right**) are good. Thus, the adapted model with an inserted SAE more closely follows the original

