# OpenReview forum: "Low-Rank Adapting Models for Sparse Autoencoders"
_ICML.cc/2025/Conference — ICML 2025 poster_

### Official Review · Reviewer_YW8j · 2025-03-11

**Overall Recommendation:** 5

**Summary:**

This paper proposes improving the use of SAEs in LLMs by fitting a LORA to the LLM with the SAE inserted. The results indicate that LORAs are an effective countermeasure to the reduction in performance of inserted SAEs.

**Claims And Evidence:**

The claims of the paper are well supported through a variety of experiments. The use of various different SAEs, LLMs, and training strategies is thorough and convincing.

**Essential References Not Discussed:**

No

**Experimental Designs Or Analyses:**

I didn't see any particular issues with the experimental designs. I was a bit confused with 5.2 in general. While I am familiar with steering LLMs, I am only vaguely able to follow the setup for this experiment. I feel Pres et al.'s setup needs to be more clearly articulated, at least in the supplement.

**Methods And Evaluation Criteria:**

The eval is in line with current work on evaluating SAEs. Everything is very sensible.

**Other Comments Or Suggestions:**

- What is "nats"? (e.g., line 088 "0.17 nats")
- captions above tables is a little unusual

**Other Strengths And Weaknesses:**

Strengths:
- the paper is well organized and easy to follow.
- the proposed method is sensible and intuitive, solving a clear problem with SAE usage
- the experiments are thorough; I cannot think of any additional work I'd want to see done

Weaknesses:
- the method is somewhat a simple premise: SAE + LORA. There isn't really a novel contribution in making LORA or SAE more amendable to solving the problem.

**Questions For Authors:**

Can you explain 5.2 in more detail?

**Relation To Broader Scientific Literature:**

This paper fits nicely into the literature of mech interpretability and is extremely up to date with the latest advancements in this fast pace field. Topk SAE is just published in ICLR and SAE bench was only announced in December.

**Theoretical Claims:**

N/A

---

> ### Author Rebuttal · Authors · 2025-04-01
>
> Thank you for your feedback and time! We are especially glad to hear you found our experiments and results to be thorough and convincing.
>
> ---
>
> > Can you explain 5.2 in more detail?
>
> Certainly! We apologize for not being clearer and hope the below will explain in more detail how we’re computing our metric:
>
> For a given dataset D, for both the base and LoRA model:
> - For a given text sample $i \in D$, we compute the average per token log likelihood $a_i$.
> - On the same text sample $i \in D$, we compute the average per token log likelihood $b_i$ when we steer with an SAE feature $f$.
> - We repeat this over a dataset of text samples.
> - We normalize all the log likelihoods (both the ones with and without steering) by subtracting a constant and scaling factor such that the non-steering log likelihoods range from 0 to 100. The constant and scaling factor are computed based on the non-steered log-likelihoods. We normalize this way so that the change in log likelihood after steering loosely represents a % change in likelihood after steering. We keep track of all the changes in LL after steering as $\delta_i$.
>
> We record $\Delta_D$ as the mean change $\delta_{i, \text{LoRA}} - \delta_{i, \text{base}}$, with standard error estimates.
>
> On a dataset consisting of text that exhibit the SAE feature (which we denote as “positive” texts), a model that steers more effectively would have a higher change in LL after steering (that is $\Delta_{\text{positive}} > 0$) because it would indicate the positive text sample is more likely after steering.
>
> However, on a dataset of texts that do not exhibit the SAE feature (which we denote as “negative” texts), we want the change in LL to remain the same or slightly decrease  (that is, $\Delta_{\text{negative}} \le 0$) after steering, as this indicates that steering with an SAE feature f does not significantly affect the likelihood of unrelated texts.
>
>
> > the method is somewhat a simple premise: SAE + LORA. There isn’t really a novel contribution in making LORA or SAE more amenable to solving the problem.
>
> We agree the method is simple! We believe that’s a strong appeal of our results, as practitioners prefer implementing simpler techniques. We also study the performance of our technique in many settings and analyze why the method works (Section 6), which we believe goes beyond merely combining SAEs and LoRA.
>
> > What is “nats”? (e.g., line 088 “0.17 nats”)
>
> nats is a unit measuring information and is the unit for cross entropy loss (it is the base e analog of bits)
>
> > captions above tables is a little unusual
>
> We agree and usually prefer having them on the bottom, but having captions above the table is required as per the ICML style guide.
>
> We are also excited to share we ran new evaluations on just our adapted model (no SAE inserted) and found that it outperforms the original model on many of the capability benchmarks across different model sizes (see table). We find it compelling that the models created using our method not only demonstrate stronger evidence of actually using the interpretable SAE features downstream, but also match or even exceed the general performance capabilities of the original pretrained model.
>
> *Italicized* metrics indicate better performing between models with SAE inserted
>
> **Bolded** metrics indicate best performing method overall.
>
> **Gemma-2-2b**
> |Metric|SAE|SAE+LoRA|Original|LoRA|
> |--------------|---------------|--------------------|----------------|------------------|
> |MMLU|44.2±0.4|*45.8±0.4*|49.3±0.4|**50.0±0.4**|
> |HellaSwag|50.9±0.5|*52.1±0.5*|55.0±0.5|**56.0±0.5**|
> |BLEU|29.9±1.6|*30.6±1.6*|30.4±1.6|**32.4±1.6**|
> |ROUGE-1|28.2±1.6|*28.5±1.6*|26.9±1.6|**30.2±1.6**|
> |ROUGE-2|24.8±1.5|*26.6±1.5*|25.6±1.5|**29.1±1.6**|
> |MC1|23.1±1.5|*23.4±1.5*|24.1±1.5|**24.3±1.5**|
>
> ---
>
> **GEMMA-2-9b**
> |**Metric**|SAE|SAE+LoRA|Original|LoRA|
> |--------------|---------------|--------------------|----------------|------------------|
> |MMLU|64.2±0.4|*65.7±0.4*|**70.0±0.4**|68.8±0.4|
> |HellaSwag|58.3±0.5|*59.6±0.5*|61.2±0.5|**61.9±0.5**|
> |BLEU|40.9±1.7|*42.4±1.7*|**43.8±1.7**|43.6±1.7|
> |ROUGE-1|39.0±1.7|*40.6±1.7*|42.7±1.7|**43.5±1.7**|
> |ROUGE-2|33.4±1.7|*36.4±1.7*|38.3±1.7|**38.8±1.7**|
> |MC1|27.1±1.6|*28.0±1.6*|30.5±1.6|**31.0±1.6**|
>
> ---
>
> **GEMMA-2-27b**
> |**Metric**|SAE|SAE+LoRA|Original|LoRA|
> |--------------|---------------|--------------------|----------------|------------------|
> |MMLU|70.9±0.4|*71.3±0.4*|72.1±0.3|**72.7±0.3**|
> |HellaSwag|61.0±0.5|*62.7±0.5*|65.3±0.5|**65.5±0.5**|
> |BLEU|*40.9±1.7*|38.9±1.7|41.1±1.7|**41.9±1.7**|
> |ROUGE-1|*41.0±1.7*|38.3±1.7|40.9±1.7|**41.7±1.7**|
> |ROUGE-2|*37.1±1.7*|35.3±1.7|36.2±1.7|**37.1±1.7**|
> |MC1|30.2±1.6|*31.5±1.6*|**33.8±1.7**|32.9±1.6|
>
> ---
>
> Thank you again for taking the time to review the paper and providing helpful feedback! Do the above actions address your concerns with the paper? And are there any further clarification or modifications we could make to improve your score?

---

> > ### Comment · Reviewer_YW8j · 2025-04-04
> >
> > Thanks authors. I like the additional experiments. I strongly believe this work should be presented and is of interest to the mechanistic interpretability community, so I will raise my score.

---

### Official Review · Reviewer_xLuu · 2025-03-14

**Overall Recommendation:** 3

**Summary:**

This paper trains some parts of a Transformer with LoRAs in order to make an SAE more accurate at reconstructing activations as a sparse linear sum. They achieve impressive upstream metric results at a low cost, and also achieve reasonable SAEBench downstream (ish!) results too.

**Claims And Evidence:**

The claims about this training process seem accurate, but there are some downstream applications I am concerned with: see my questions.

Additionally, I am concerned that far too much focus is on improving simple SAE metrics rather than improving interpretability concretely. This paper is very long anyway, so the authors may have considered it out of scope, but I don't think there evidence on circuit analysis is anywhere near sufficient to suggest that their technique is likely useful for improving our understanding of models.

At absolute minimum, the authors should be much more open about the limitations of their work.

**Essential References Not Discussed:**

N/A this paper is comprehensive with references.

**Experimental Designs Or Analyses:**

The experimental setup is done well.

**Methods And Evaluation Criteria:**

The evaluation is fairly comprehensive, e.g. SAEBench and also normal CE loss style evals

I was somewhat sad that no novel findings were made to advance interp -- e.g. a circuit analysis experiment that seem to go a lot better than expected with these new tools.

**Other Comments Or Suggestions:**

"Concretely, for each frozen weight matrix..." -- surely this is supposed to say **un**-frozen?

**Other Strengths And Weaknesses:**

N/A

**Questions For Authors:**

1a. Why is this a promising method for circuit discovery? I see significant downside. Normally, you can take off-the-shelf SAEs, and choose a subset of them to use for circuit analysis. E.g. https://openreview.net/forum?id=sdLwJTtKpM off the top of my head. Your paper requires a very expensive training procedure that will need to be redone if different SAEs need to be attached.

1b. In addition to those worries about cirucit analysis, I don't think you can use error terms with your approach? But often attaching lots of SAEs is very lossy, so errors terms are essential.

2. Why is this a promising method for applications of probing? A key reason to do probing work, from my perspective, is to be able to make monitors production-ready systems, e.g. frontier models at labs. But if the way to monitor requires editing weights before the
SAE then the latents will not be able to used on their own as probes (for example), entirely new model weights will need be used. And it seems your weights have a decent dip in performance on e.g. MMLU so this seems to limit applications of your tool

**Relation To Broader Scientific Literature:**

Interpretability has wide appeal and I think if this research enabled research (unclear) it would be impactful in wider communities.

**Theoretical Claims:**

Theoretical claims were fine.

---

> ### Author Rebuttal · Authors · 2025-04-01
>
> We are greatly thankful for your time and feedback, especially related to the limitations of our work. We were glad to hear you found the evaluations comprehensive.
>
> ---
>
> > Additionally, I am concerned that far too much focus is on improving simple SAE metrics… At absolute minimum, the authors should be much more open about the limitations of their work.
>
> This is a fair criticism, thank you! We agree that our multi-SAE section does not directly show improvements to SAE circuits. We have added the following to the limitations section, please let us know what you think!
> ```
> While our approach develops models that more effectively leverage the interpretable features learned by SAEs in downstream tasks, we do not yet show that these can be leveraged into additional insights as to how the original model internally computes using these features. We leave this important direction for future work.
> ```
>
> > In addition to those worries about circuit analysis, I don’t think you can use error terms with your approach? But often attaching lots of SAEs is very lossy, so error terms are essential.
>
> This is a good question! It is true that one cannot use error terms with our method to obtain the activations of the original model, but in return we get the benefit that our method directly attacks the issue that “attaching lots of SAEs is very lossy.” For example, note that in Figure 6, even inserting 7 SAEs with our method results in a reasonable cross entropy loss of 2.78, and inserting 3 SAEs into the LoRA model has better loss than the pretrained model with 1 SAE.
>
> We also note that one could use error terms to recover the performance of the adapted model, which we show below is similar (and in some cases even better) than the original model.
>
> > Your paper requires a very expensive training procedure that will need to be redone if different SAEs need to be attached
>
> While it’s true the training procedure needs to be redone if different SAEs are attached, the training procedure is actually very cheap. Taking off the shelf SAEs, training LoRA adapters takes only a few minutes or hours (depending on your model size), which is many times cheaper than the original SAE training process.
>
> > Why is this a promising method for applications of probing? …if the way to monitor requires editing weights before the SAE then the latents will not be able to used on their own as probes
>
> This is a great point and something we overlooked. We have added a discussion of this to Section 5.1.
>
> However, as you note someone using our method for probing could run the adapted model without the SAE (because they would just be using it for probing), and we are excited to show that in new experiments, the adapted model with no SAE is actually as good or better than the original model on many capability benchmarks:
>
> *Italicized* indicates best performing between models with SAE inserted
>
> **Bolded** indicates best overall.
>
> **Gemma-2-2b**
> |Metric|SAE|SAE+LoRA|Original|LoRA|
> |--------------|---------------|--------------------|----------------|------------------|
> |MMLU|44.2±0.4|*45.8±0.4*|49.3±0.4|**50.0±0.4**|
> |HellaSwag|50.9±0.5|*52.1±0.5*|55.0±0.5|**56.0±0.5**|
> |BLEU|29.9±1.6|*30.6±1.6*|30.4±1.6|**32.4±1.6**|
> |ROUGE-1|28.2±1.6|*28.5±1.6*|26.9±1.6|**30.2±1.6**|
> |ROUGE-2|24.8±1.5|*26.6±1.5*|25.6±1.5|**29.1±1.6**|
> |MC1|23.1±1.5|*23.4±1.5*|24.1±1.5|**24.3±1.5**|
>
> ---
>
> **GEMMA-2-9b**
> |**Metric**|SAE|SAE+LoRA|Original|LoRA|
> |--------------|---------------|--------------------|----------------|------------------|
> |MMLU|64.2±0.4|*65.7±0.4*|**70.0±0.4**|68.8±0.4|
> |HellaSwag|58.3±0.5|*59.6±0.5*|61.2±0.5|**61.9±0.5**|
> |BLEU|40.9±1.7|*42.4±1.7*|**43.8±1.7**|43.6±1.7|
> |ROUGE-1|39.0±1.7|*40.6±1.7*|42.7±1.7|**43.5±1.7**|
> |ROUGE-2|33.4±1.7|*36.4±1.7*|38.3±1.7|**38.8±1.7**|
> |MC1|27.1±1.6|*28.0±1.6*|30.5±1.6|**31.0±1.6**|
>
> ---
>
> **GEMMA-2-27b**
> |**Metric**|SAE|SAE+LoRA|Original|LoRA|
> |--------------|---------------|--------------------|----------------|------------------|
> |MMLU|70.9±0.4|*71.3±0.4*|72.1±0.3|**72.7±0.3**|
> |HellaSwag|61.0±0.5|*62.7±0.5*|65.3±0.5|**65.5±0.5**|
> |BLEU|*40.9±1.7*|38.9±1.7|41.1±1.7|**41.9±1.7**|
> |ROUGE-1|*41.0±1.7*|38.3±1.7|40.9±1.7|**41.7±1.7**|
> |ROUGE-2|*37.1±1.7*|35.3±1.7|36.2±1.7|**37.1±1.7**|
> |MC1|30.2±1.6|*31.5±1.6*|**33.8±1.7**|32.9±1.6|
>
>
>
> > …surely this is supposed to say un-frozen?
>
> We think this is written correctly. To clarify, in LoRA finetuning, the weight matrices W of the original model are frozen, and only low rank matrices A and B are learned. The forward pass through the model is then (W + AB)x instead of Wx. We hope this clarifies things; we apologize for not being more clear.
>
> ---
>
> Thank you again for taking the time to review the paper and providing helpful feedback! Do the above actions address your concerns with the paper? If not, what further clarification or modifications could we make to improve your score?

---

> > ### Comment · Reviewer_xLuu · 2025-04-04
> >
> > > We have added the following to the limitations section
> >
> > I broadly like that comment but think it should explicitly have "circuit" as a substring somewhere. And reference with `\Cref` the circuits section.
> >
> > > We think this is written correctly
> >
> > Yes, I follow now, thanks
> >
> > Staring more at the metrics, I do agree there is not too much of a capability hit to Gemma 27B which is the most relevant model. So I update my score to a 3 (weak accept).

---

### Official Review · Reviewer_rL2B · 2025-03-14

**Overall Recommendation:** 3

**Summary:**

This paper introduces an approach for improving sparse autoencoders (SAEs) used in language model interpretability by using Low-Rank Adaptation (LoRA) to finetune the language model itself around a previously trained SAE. Their method freezes both the original model and the SAE while only training low-rank adapters to minimize the KL divergence between the model's original outputs and the outputs with the SAE inserted.
The authors present experiments showing their approach reduces the cross entropy loss gap by 30-55% across various settings (SAE sparsity, width, model size, LoRA rank, and model layer). They demonstrate computational efficiency compared to end-to-end SAEs and claim improvements on downstream tasks, including SAEBench metrics and general language model capabilities.

**Claims And Evidence:**

While the paper presents comprehensive experimental evidence for its technical claims, there is a fundamental conceptual issue that undermines the paper's premise:
1. The paper claims to improve model interpretability by adapting the model to better accommodate SAEs. However, this approach appears to misunderstand the purpose of SAEs in mechanistic interpretability. SAEs are meant to be analytical tools that reveal the features present in a fixed, pretrained model's activations. By modifying the model itself to better accommodate the SAE, the authors are effectively changing what's being analyzed rather than improving the analysis method.
2. The cross-entropy loss improvements are well-documented, but it's unclear whether these improvements actually serve the goal of better interpretability or simply represent overfitting the model to better accommodate a specific analysis tool.

**Essential References Not Discussed:**

N/A

**Experimental Designs Or Analyses:**

The experiments technically demonstrate what they claim, but several issues arise:

1. There's no evaluation of whether the features discovered after model adaptation are actually more interpretable to humans, only that they perform better on automated metrics.
2. The paper doesn't address whether adapting the model might actually hide or modify important original features that would be relevant for interpretability.
3. The feature steering experiments don't clearly show that the steered features are more semantically coherent, only that they have stronger statistical effects.

**Methods And Evaluation Criteria:**

The proposed methods are technically sound but conceptually problematic for the interpretability goal:

1. In mechanistic interpretability, the model being analyzed should remain fixed - modifying it defeats the purpose of understanding the original model's behavior.
2. While the authors use SAEBench metrics to claim improved interpretability, it's unclear whether these metrics actually measure feature interpretability or just the degree to which the model has been tuned to make the SAE perform better.
3. The evaluation focuses heavily on cross-entropy loss and general model performance metrics, which are not necessarily aligned with the goal of finding more interpretable features.

**Other Comments Or Suggestions:**

What does the $\textbf{f}$ in Eq. 5 (line 158)  stand for?

**Other Strengths And Weaknesses:**

Strengths:

1. The technical implementation is sound and well-executed.
2. The experiments are comprehensive across different model scales and SAE configurations.
3. The method does effectively reduce the computational cost compared to e2e SAEs.

Weaknesses:

1. The central premise appears to fundamentally misunderstand the purpose of SAEs in mechanistic interpretability. SAEs are meant to discover interpretable features in fixed models, not to guide model adaptation.
2. By modifying the model to better fit the SAE, the authors may be creating an artificial scenario that doesn't reflect how the original model actually processes information, undermining the core goal of interpretability.
3. The paper conflates sparsity with interpretability. While sparsity can help with interpretability, the primary goal is to find semantically meaningful features. The paper focuses almost exclusively on improving sparse reconstruction without demonstrating improved semantic interpretability.
4. The approach may actually hinder true interpretability by adapting the model to make the SAE work better, potentially masking or altering the original computational mechanisms that interpretability research aims to discover.
5. The paper doesn't provide human evaluation of the interpretability of the features found after model adaptation compared to those from standard SAEs.

**Questions For Authors:**

Questions for Authors

1. How do you reconcile your approach of modifying the model with the traditional goal of mechanistic interpretability, which is to understand the computational mechanisms of fixed, pretrained models? Does changing the model not defeat the purpose of analyzing it?

2. Do you have evidence that the features discovered after model adaptation are more semantically interpretable to humans, rather than just achieving better reconstruction or downstream metrics?

3. Have you analyzed whether the adapted model still exhibits the same internal computational patterns as the original model? If not, how can we be confident that we are still studying the same phenomena?

4. Your method essentially changes what is being analyzed rather than improving the analysis method. How do you ensure that the modified model still maintains the properties that made the original model worth studying?

5. Sparsity is typically used as a proxy for interpretability, not as the end goal. Your paper focuses heavily on improving the sparsity-performance tradeoff, but does this actually translate to more humanly interpretable features? What evidence do you have for this?

**Relation To Broader Scientific Literature:**

The paper situates itself within mechanistic interpretability literature but appears to misunderstand a fundamental aspect of this field. Traditional mechanistic interpretability approaches aim to analyze fixed models to understand how they work internally. By modifying the model to better accommodate the analysis tool (SAE), this paper reverses the relationship between the object of study and the analytical method.
This reversal represents a significant departure from the goals described in foundational works and more recent SAE application papers, which focus on understanding fixed models rather than adapting models to analysis methods.

**Theoretical Claims:**

There is no theoretical claims in this paper.

---

> ### Author Rebuttal · Authors · 2025-04-01
>
> We are grateful for your time and thorough feedback, especially with regards to our methodology. We were glad to hear you found the implementation to be well executed.
>
> ---
>
> > How do you reconcile your approach of modifying the model with the traditional goal of mechanistic interpretability?
>
> This is an important point, thank you! As we state in our limitations (Section 7.1), our method is not appropriate when the model cannot be modified. However, while much interpretability research focuses on fixed models, other work builds interpretable models from scratch or modifies existing models to be interpretable. As cited in our related work, this includes Lai & Heimersheim, 2024; Lai & Huang, 2024; Elhage et al., 2022b; Liu et al., 2023; Liu et al., 2024; Heimersheim 2024. We consider our paper well situated in this latter category and thus an important research contribution.
>
> Moreover, because pretrained models perform poorly with SAEs inserted, one may be skeptical if models are actually using the more interpretable SAE latents. An interpretable basis is unhelpful if the model does not actually use it. Because our method optimizes the model to use SAE latents downstream, we can be more confident they are really using them.
>
> > Have you analyzed whether the adapted model still exhibits the same internal computational patterns?
>
> Great question! In Figure 11, we compare the distance between the downstream layer activations of the pretrained model (gemma-2-2b) with the downstream layer activations of 1) the pretrained model with the SAE, and 2) the adapted model with the SAE. We find that the adapted model with the SAE has a higher cosine similarity with the pretrained model’s original activations than the pretrained model with the SAE. Thus, our adapted model + SAE is more similar to the original model than the pretrained model + SAE.
>
> We also ran a quick experiment comparing just the adapted and pretrained model (no SAE involved). We found the average per-layer cosine similarity between pretrained and adapted model activations on the Pile was consistently greater than 0.99 (for comparison, the average cosine similarity with the inserted SAE is ~0.95).
>
> > How do you ensure that the modified model still maintains the properties that made the original model worth studying?
>
> Great question! It is true that by modifying the model, we may be losing capabilities of the original model. To allay this concern, we ran our adapted model on the benchmarks from Table 5 and found that not only does the modified model still perform competitively with the pretrained model, but that in many cases it *outperforms* the original pretrained model.
>
> *Italicized* metrics indicate best performing between models with SAE inserted
>
> **Bolded** metrics indicate best method overall.
>
> **Gemma-2-2b**
> |Metric|SAE|SAE+LoRA|Original|LoRA|
> |--------------|---------------|--------------------|----------------|------------------|
> |MMLU|44.2±0.4|*45.8±0.4*|49.3±0.4|**50.0±0.4**|
> |HellaSwag|50.9±0.5|*52.1±0.5*|55.0±0.5|**56.0±0.5**|
> |BLEU|29.9±1.6|*30.6±1.6*|30.4±1.6|**32.4±1.6**|
> |ROUGE-1|28.2±1.6|*28.5±1.6*|26.9±1.6|**30.2±1.6**|
> |ROUGE-2|24.8±1.5|*26.6±1.5*|25.6±1.5|**29.1±1.6**|
> |MC1|23.1±1.5|*23.4±1.5*|24.1±1.5|**24.3±1.5**|
>
> ---
>
> **GEMMA-2-9b**
> |**Metric**|SAE|SAE+LoRA|Original|LoRA|
> |--------------|---------------|--------------------|----------------|------------------|
> |MMLU|64.2±0.4|*65.7±0.4*|**70.0±0.4**|68.8±0.4|
> |HellaSwag|58.3±0.5|*59.6±0.5*|61.2±0.5|**61.9±0.5**|
> |BLEU|40.9±1.7|*42.4±1.7*|**43.8±1.7**|43.6±1.7|
> |ROUGE-1|39.0±1.7|*40.6±1.7*|42.7±1.7|**43.5±1.7**|
> |ROUGE-2|33.4±1.7|*36.4±1.7*|38.3±1.7|**38.8±1.7**|
> |MC1|27.1±1.6|*28.0±1.6*|30.5±1.6|**31.0±1.6**|
>
> ---
>
> **GEMMA-2-27b**
> |**Metric**|SAE|SAE+LoRA|Original|LoRA|
> |--------------|---------------|--------------------|----------------|------------------|
> |MMLU|70.9±0.4|*71.3±0.4*|72.1±0.3|**72.7±0.3**|
> |HellaSwag|61.0±0.5|*62.7±0.5*|65.3±0.5|**65.5±0.5**|
> |BLEU|*40.9±1.7*|38.9±1.7|41.1±1.7|**41.9±1.7**|
> |ROUGE-1|*41.0±1.7*|38.3±1.7|40.9±1.7|**41.7±1.7**|
> |ROUGE-2|*37.1±1.7*|35.3±1.7|36.2±1.7|**37.1±1.7**|
> |MC1|30.2±1.6|*31.5±1.6*|**33.8±1.7**|32.9±1.6|
>
>
>
> > Do you have evidence that the features discovered after model adaptation are more semantically interpretable to humans?
>
> Thank you for this question; we apologize for not being clearer! We are not trying to find more interpretable features. Indeed, in sections 4.1 & 4.2 (where we recover up to 55% of the loss gap), we adapt only after the SAE, so the features (and their interpretability) are *unchanged*. Rather, we are creating an alternative model that better uses these interpretable features downstream.
>
> > What does the f in Eq. 5 (line 158) stand for?
>
> SAE dictionary **f**eatures.
>
> ---
>
> Thank you again for taking the time to review the paper and providing helpful feedback! Do the above actions address your concerns with the paper? If not, what further clarification or modifications could we make to improve your score?

---

### Official Review · Reviewer_PHUR · 2025-03-15

**Overall Recommendation:** 4

**Summary:**

This paper introduces a novel approach to improve the interpretability of SAE by using LoRA to fine-tune **LLMs** around previously trained SAEs. Unlike previous work that focused on optimizing SAE architectures, this approach optimizes the language model itself to work better with an existing SAE. Across various experiments with SAE sparsity, width, language model size, LoRA rank, and model layer, the method reduces the cross entropy loss gap by 30% to 55% when SAEs are inserted during the forward pass. The technique can adapt multiple SAEs simultaneously, significantly reducing compound cross entropy loss (e.g., from 7.83 to 2.78 nats with 7 SAEs). The paper demonstrates that improving model interpretability is not limited to post-hoc SAE training; Pareto improvements can also be achieved by directly optimizing the model while keeping the SAE fixed.

**Claims And Evidence:**

Mostly. see comments below.

**Essential References Not Discussed:**

No

**Experimental Designs Or Analyses:**

Overall, the experiment setup is mostly comprehensive and sound, but there are still points to be improved. More ablation studies and analyses could be conducted to prove that the performance gain of the method does not randomly appear but is a constant improvement. For example:

- More diverse model architectures could be tested, e.g., Mistral and Qwen (where there are already some SAEs and there's no need to train SAE from zero).
- Different SAE sizes and capabilities could also be considered to prove that this method is constantly surpassing SAE rather than needing a strong SAE to start.

**Methods And Evaluation Criteria:**

The idea of training the model according to interpretability methods is very novel, and I consider it to be the main reason for the paper to be accepted. The evaluation criteria is frontier and up-to-date.

**Other Comments Or Suggestions:**

No

**Other Strengths And Weaknesses:**

The significance and scalability of this method could be a little concerning, and I also quite doubt whether this method could actually be applied to real-world models (to address this, maybe the authors could test the method on benchmarks that is closer to real-world usage, but that won't affect my general evaluation of this idea).

**Questions For Authors:**

1. Could you replicate the performance experiment on standard NLP benchmarks on the 27B model? I'm concerned about whether this method could scale up to bigger models (and also more complicated features and capabilities).

**Relation To Broader Scientific Literature:**

Yes, the relation is clearly identified by the authors, and the method is indeed improving the performance and interoperability compared to related works and has solved some of the concerns (e.g., training time and efficiency) proposed in previous works.

**Theoretical Claims:**

No theoretical claims have been made in this paper.

---

> ### Author Rebuttal · Authors · 2025-04-01
>
> We thank you for your time and help, especially in regards to your suggestions for additional experiments, which we have now run. We are especially glad that you appreciated the novelty of modifying the model itself to better fit a sparse autoencoder, as we agree that this idea is the major impact of our work.
>
> ---
>
> > More diverse model architectures could be tested, e.g., Mistral and Qwen (where there are already some SAEs and there's no need to train SAE from zero).
>
> Thank you for this suggestion! We ran additional experiments on the layer 16 Mistral 7B SAE from SAELens and found that our method still works (we did not see a Qwen SAE on SAELens). The Mistral 7B base model has a CE loss of 1.5485 on our val set, the model with the SAE inserted has a CE loss of 1.8007, and the LoRA model with SAE inserted has a CE loss of 1.6548, reducing the cross entropy loss gap by 57.8%.
>
> > Different SAE sizes and capabilities could also be considered to prove that this method is constantly surpassing SAE rather than needing a strong SAE to start.
>
> We show that our method works on a large number of SAE sizes and sparsities in Section 4; for all of them it surpasses the performance of the original SAE. We also show that even for partially trained SAEs our method works well; see Figure 1. Please let us know if there is anything we could add to the main text (experiments or wording) to make this stronger, thank you!
>
>
> > Could you replicate the performance experiment on standard NLP benchmarks on the 27B model? I'm concerned about whether this method could scale up to bigger models (and also more complicated features and capabilities).
>
> Thank you for this suggestion! Below, we include a version of Table 5 with a 27B model. Our method seems to scale up well to bigger models. Interestingly, the adapted model alone outperforms the original pretrained model on many of the capability benchmarks across all model sizes; we believe this is evidence our method scales well, and that the adapted models from our method could be of interest for real world use cases.
>
> *Italicized* metrics indicate best performing between models with SAE inserted
> **Bolded** metrics indicate best performing method overall.
>
> **Gemma-2-2b**
> |Metric|SAE|SAE+LoRA|Original|LoRA|
> |--------------|---------------|--------------------|----------------|------------------|
> |MMLU|44.2±0.4|*45.8±0.4*|49.3±0.4|**50.0±0.4**|
> |HellaSwag|50.9±0.5|*52.1±0.5*|55.0±0.5|**56.0±0.5**|
> |BLEU|29.9±1.6|*30.6±1.6*|30.4±1.6|**32.4±1.6**|
> |ROUGE-1|28.2±1.6|*28.5±1.6*|26.9±1.6|**30.2±1.6**|
> |ROUGE-2|24.8±1.5|*26.6±1.5*|25.6±1.5|**29.1±1.6**|
> |MC1|23.1±1.5|*23.4±1.5*|24.1±1.5|**24.3±1.5**|
>
> ---
>
> **GEMMA-2-9b**
> |**Metric**|SAE|SAE+LoRA|Original|LoRA|
> |--------------|---------------|--------------------|----------------|------------------|
> |MMLU|64.2±0.4|*65.7±0.4*|**70.0±0.4**|68.8±0.4|
> |HellaSwag|58.3±0.5|*59.6±0.5*|61.2±0.5|**61.9±0.5**|
> |BLEU|40.9±1.7|*42.4±1.7*|**43.8±1.7**|43.6±1.7|
> |ROUGE-1|39.0±1.7|*40.6±1.7*|42.7±1.7|**43.5±1.7**|
> |ROUGE-2|33.4±1.7|*36.4±1.7*|38.3±1.7|**38.8±1.7**|
> |MC1|27.1±1.6|*28.0±1.6*|30.5±1.6|**31.0±1.6**|
>
> ---
>
> **GEMMA-2-27b**
> |**Metric**|SAE|SAE+LoRA|Original|LoRA|
> |--------------|---------------|--------------------|----------------|------------------|
> |MMLU|70.9±0.4|*71.3±0.4*|72.1±0.3|**72.7±0.3**|
> |HellaSwag|61.0±0.5|*62.7±0.5*|65.3±0.5|**65.5±0.5**|
> |BLEU|*40.9±1.7*|38.9±1.7|41.1±1.7|**41.9±1.7**|
> |ROUGE-1|*41.0±1.7*|38.3±1.7|40.9±1.7|**41.7±1.7**|
> |ROUGE-2|*37.1±1.7*|35.3±1.7|36.2±1.7|**37.1±1.7**|
> |MC1|30.2±1.6|*31.5±1.6*|**33.8±1.7**|32.9±1.6|
>
>
> ---
>
> Thank you again for taking the time to review the paper and providing helpful feedback! Do the above actions address your concerns with the paper? If not, what further clarification or modifications could we make to improve your score?

---

> > ### Comment · Reviewer_PHUR · 2025-04-05
> >
> > Thanks for your rebuttal! I've raised my score accordingly.

---

### Decision · Program_Chairs · 2025-05-01

**Decision:**

Accept (poster)

**Comment:**

This paper proposes using Low-Rank Adaptation (LoRA) to finetune language models around previously trained Sparse Autoencoders (SAEs), aiming to improve model interpretability. The authors demonstrate that their approach reduces cross entropy loss gaps by 30-55% when SAEs are inserted during the forward pass, while being computationally more efficient than end-to-end SAE training.

The reviewers found the paper to be well-executed technically, with comprehensive experiments across different model scales and SAE configurations. They appreciated the thorough evaluation using both standard metrics and SAEBench. While some concerns were raised about the conceptual alignment with traditional mechanistic interpretability goals and the need for more evidence of improved semantic interpretability, the authors provided detailed responses and additional experiments that largely addressed these issues.

The paper makes a meaningful contribution to the growing body of work on building more interpretable models, even if the core method is relatively simple. The additional experiments showing maintained or improved model capabilities after adaptation were particularly compelling. The clear writing, thorough empirical evaluation, and practical utility of the method were consistently highlighted by reviewers.

The revised version is expected to incorporate expanded discussion of limitations, particularly regarding circuit analysis applications, and provide clearer explanation of the steering experiments. With these modifications, the paper will serve as a valuable contribution to the field of model interpretability.